# Evaluation of Three Carbapenemase-Phenotypic Detection Methods and Emergence of Diverse VIM and GES Variants among *Pseudomonas aeruginosa* Isolates in Tunisia

**DOI:** 10.3390/antibiotics11070858

**Published:** 2022-06-27

**Authors:** Sana Ferjani, Elaa Maamar, Asma Ferjani, Lamia Kanzari, Ilhem Boutiba Ben Boubaker

**Affiliations:** 1Faculty of Medicine of Tunis, University of Tunis El Manar, LR99ES09, Tunis 1007, Tunisia; alaa_maamar@hotmail.com (E.M.); aferjani76@gmail.com (A.F.); lamiakanzari@yahoo.fr (L.K.); ilhem.boutiba@gmail.com (I.B.B.B.); 2Laboratory of Microbiology, Charles Nicolle Hospital, Tunis 1006, Tunisia

**Keywords:** *Pseudomonas*, GES, VIM, ICU

## Abstract

Background: Since 2012, few reports on the molecular epidemiology of *Pseudomonas aeruginosa* were reported in Tunisia. Objectives: This study aimed to evaluate carbapenem-resistance determinants and molecular epidemiology and to compare the carbapenemase-phenotypic detection methods of multidrug-resistant *P. aeruginosa* isolates. Methods: During a period of four years (2014 to 2017), all imipenem-ceftazidime-resistant *P. aeruginosa* isolates were retrospectively selected at the microbial laboratory of Charles Nicolle hospital of Tunis. These isolates were examined by the modified Hodge test, modified carbapenem inactivation method (mCIM), and another mCIM, called CIMTris, and their performance was evaluated using PCR analysis as the gold standard. Results: A total of 35 isolates were recovered among patients hospitalized in different units. All strains were colistin-susceptible.All carbapenem-resistant isolates showed a high-level resistance to carbapenems. CIMTris and mCIM showed 96.15% and 46.15% sensitivity and 44.44% and 100% specificity, respectively, for detecting carbapenemase production.Conclusions: CIMTris is a promising approach for detecting carbapenemase activity in *P. aeruginosa* and merits further testing. Moreover, this study described the first detection of GES-5- and GES-9-producing *P. aeruginosa* in Tunisia as well as the co-occurrence of the *bla*_GES-5_ and *bla*_VIM-11_ carbapenemase genes in one isolate. These findings are of great concern because the rapid dissemination of MDR strains represents a major therapeutic and epidemiological threat.

## 1. Introduction

*Pseudomonas aeruginosa* is an important pathogen that causes various opportunistic and acute nosocomial-acquired infections, especially in immunocompromised patients. Its high intrinsic resistance and ability to develop multidrug resistance produce serious therapeutic problems. Carbapenems have been considered first-line agents to treat severe cases of *P. aeruginosa* infections [1]. Resistance to carbapenems can stem from the production of carbapenemases or other mechanisms such as mutation in the *oprD* gene, the overproduction of cephalosporinases, the over-expression of effluxpumps, or combinations of these mechanisms [2].

The most commonly reported carbapenemases among *P. aeruginosa* are metallo-β-lactamases (MBLs) (e.g., the Verona Imipenemase (VIM), Imipenemase (IMP), Sao Paulo metallo enzyme (SPM), German imipenemase (GIM), and New Dehli metallo β-lactamase (NDM) types) and, to a lesser extent, Ambler class A carbapenemases (e.g., *Klebsiella pneumoniae* carbapenemase (KPC) and some Guyana extended-spectrum (GES)-type enzymes [3]. MBL enzymes are also able to hydrolyze penicillin and cephalosporins [4,5]. Moreover, the MBL VIM-2, initially reported in France [4], has emerged and has been reported to be the main MBL determinant in *P. aeruginosa* isolates in Tunisia and worldwide during the past two decades [6,7,8,9]. Their increase in recent years, which is associated with high mortality, morbidity, long hospital stays, and increased costs, emphasizes the need for the detection of these isolates to avoid therapeutic failures and nosocomial outbreaks [8,9].

It should be noted that detecting carbapenemases is more difficult in *P. aeruginosa* compared to *Enterobacteriaceae* [10]. Currently, several phenotypic methods are available, but most of them are unsuitable for clinical laboratories to perform on a routine basis [10,11]. Thus, standardized carbapenemase detection methods using routine phenotypic screening tests are still controversial.

Since 2012, few reports on the molecular epidemiology of *P. aeruginosa* were reported in Tunisia [6]. The aim of our study was, therefore, to evaluate changes in carbapenem-resistance determinants and molecular epidemiology and to compare carbapenemase-phenotypic detection methods of multidrug-resistant *P. aeruginosa* isolates recovered at Charles Nicolle Hospital of Tunis, Tunisia, during a 4-year period.

## 2. Materials and Methods

### 2.1. Bacterial Isolates

A total of 80 imipenem–ceftazidime-resistant *P. aeruginosa* (ICRPA) isolates were retrospectively selected from the frozen stocks kept in brain heart infusion with 20% glycerol at −80 °C at the microbial laboratory of Charles Nicolle hospital of Tunis during a period of four years (2014 to 2017).

From frozen stock, each isolate was subcultured twice on tryptic soy agar (Biorad, Marnesta la Coquette, France), incubating each subculture in ambient air at 35 °C ± 2 °C for 18 to 24 h to ensure purity and viability. Thus, only 35 non-duplicated clinical ICRPA isolates were successfully subcultured and included in this study.

Bacterial identification was performed using the API 20NE system (bioMérieux, Marcy l’Etoile, France).

### 2.2. Antimicrobial Susceptibility Testing

Antibiotic susceptibility testing to 16 antibiotics (ticarcillin, ticarcillin-clavulanic acid, piperacillin, piperacillin–tazobactam, aztreonam, ceftazidime, cefepime, imipenem, meropenem, gentamicin, tobramycin, amikacin, netilmicin, ciprofloxacin, levofloxacin, and fosfomicin) was performed by the agar disk diffusion method on Mueller–Hinton (MH) agar plates (Bio-Rad, Marnesta la Coquette, France), according to the CA-SFM guidelines (http://www.sfm-microbiologie.org/, accessed on 1 January 2017).

The minimum inhibitory concentrations (MICs) of imipenem and meropenem were determined by the agar dilution method according to the Clinical and Laboratory Standards Institute (CLSI) guidelines [M100-S25]. Colistin MICs were determined by the broth microdilution method using a commercialized kit (UMIC, Biocentric, Bandol-France).

*Escherichia coli* ATCC 25922 and *P. aeruginosa* ATCC 27853 were used as quality control strains in antimicrobial susceptibility testing and MICs.

### 2.3. Phenotypic Detection of Carbapenemase Production

The phenotypic detection of carbapenemases was performed by a modified Hodge test, modified carbapenem inactivation method (mCIM), and another mCIM, called CIMTris, which uses 0.5 M Tris-HCl buffer rather than water for extraction, as previously reported [10]. All carbapenemase phenotypic methods were assessed twice by different raters.

The performance of the carbapenemase phenotypic tests was evaluated using PCR analysis for *bla*_VIM_ and *bla*_GES_ as the gold standard (Table 1) [12]. Phenotypic method sensitivities and specificities were calculated according to Ilstrup [12,13].

### 2.4. Detection and Characterization of Beta-Lactamase Genes

The molecular detection of carbapenemase-encoding genes (*bla*_GES_, *bla*_KPC_, *bla*_OXA-48_, *bla*_IMP_, *bla*_VIM_, *bla*_NDM_, *bla*_SPM_, *bla*_BIC_, *bla*_AIM_, *bla*_GIM_, *bla*_SIM_, and *bla*_DIM_) was performedby PCR with previously reported conditions (Table 1) [14,15]. All PCR products were sequenced using a DNAsequencer (ABI PRISM 3130; Applied Biosystems, Foster City, CA, USA) [16].

## 3. Results

The 35 isolates had been obtained among patients hospitalized in different units (intensive care units, 63%; surgery ward, 26%; urology, 6%; and external consultation, 6%) (Figure 1) and from different types of samples (Figure 2) (low respiratory samples, 46%; pus, 14%; urine culture, 11%; blood culture, 11%; rectal samples, 9%; and material, 6%).

All strains were colistin-susceptible (MIC range 1–4 µg/mL) and were resistant to gentamicin (91%), tobramycin (91%), netilmicin (100%), amikacin (83%), ciprofloxacin (94%), and fosfomycin (100%) (Figure 3). All carbapenem-resistant isolates showed a high-level resistance to carbapenems. The MIC ranges of imipenem and meropenem were 4–512 μg/mL and 4–256 μg/mL, respectively.

Carbapenemase-encoding genes were detected in 26 strains (74%) and were identified as: *bla*_GES-5_ (n = 14), *bla*_GES-9_ (n = 2), *bla*_VIM-1_ (n = 1), *bla*_VIM-2_ (n = 9), and *bla*_VIM-11_ (n = 1). The association between *bla*_GES_ and *bla*_VIM_ was found in two strains (S15) (Table 2). None of the strains harbored the genes *bla*_KPC_, *bla*_IMP_, *bla*_OXA-48_, *bla*_SPM_, *bla*_NDM_, *bla*_BIC_, *bla*_AIM_, *bla*_GIM_, *bla*_SIM,_ and *bla*_DIM_.

Of the 35 ICRPA strains tested, 26 (74.2%) harbored acquired carbapenemase-encoding genes, and 24 of these were positive on CIMTris. Four of the nineisolates not harboring acquired carbapenemase genes were negative on CIMTris, whereas for the remaining fivewere positive. Thus, CIMTris showed 96.15% sensitivity and 44.44% specificity for detecting carbapenemase production.

The testing of mCIM in ICRPA isolates showed that 12 of 26 (46.15%) isolates harboring carbapenemase genes were positive on mCIM. All the 14 mCIM-negative isolates harbored acquired carbapenemase genes. Nine of the nine(100%) isolates not harboring acquired carbapenemase genes were negative on mCIM. Thus, mCIM showed a sensitivity of 46.15% and a specificity of 100% for detecting carbapenemase activities (Table 3 and Table 4).

## 4. Discussion

In our study, most of the collected ICRPA strains were isolated from low respiratory samples in ICU patients, confirming the data of a previous study [17]. It has been reported that most of the nosocomial infections caused by carbapenemase-producing *P. aeruginosa* (CPPA)most frequently affect patients with pneumonia associated with mechanical ventilation, and this is the main cause of chronic respiratory infection in immunocompromised patients.

Our study revealed that all isolated ICRPA remained susceptible only to colistin, indicating the dissemination of multidrug-resistant (MDR) strains and an emerging problem in our hospital. The problem of bacterial resistance to commonly used antibiotics is worldwide [2,5,8,17]. The management of ICRPA infections represents a difficult therapeutic challenge due to the increasing resistance levels of these organisms to most classes of antimicrobial agents.

The acquisition of carbapenemase genes by *P. aeruginosa* is an important cause of MDR, and therefore, the rapid and correct detection of carbapenemases is crucial [2]. Molecular methods are the gold standard in the identification of carbapenemase-producing strains, but phenotypic methods have been developed due to the high cost of molecular methods and their inability to detect new carbapenemase genes [2]. However, some phenotypic assays are still not accepted for non-fermentative Gram-negative bacilli [10]. In this study, we evaluate the performance of three phenotypic methods in the detection of carbapenemase-producing *P.aeruginosa,* including the modified Hodge test, mCIM, and CIMTris. The CIMTris differed from the mCIM by the Tris-HCl buffer that was used instead of water during the MEM inactivation step. The CIMTris showed markedly higher sensitivity than the mCIM and modified Hodge test (96.1% vs. 46.1 and 12.5%). The Tris-HCl buffer used in the CIMTris seemed to effectively extract carbapenemases of class A and B. However, the specificity of the CIMTris was lower than the other two methods.This could probably be explained by the degradation of meropenem by other carbapenemases that werenot detected in this study.Our results show that CIMTris is useful, simple, and accessible to clinical laboratories for detecting carbapenemase production in ICRPA, but PCR is needed to confirm the presence of carbapenemase-encoding genes, as previously reported [10].

Of the 35 ICRPA strains, 26 were harboring carbapenemase-encoding genes, and 16 strains were carrying class A beta-lactamases. The coexistence of *bla*_GES-5_ and *bla*_VIM-11_ was found in one strain. Thus, our results show a predominance of *bla*_GES-5_, which is not in agreement with earlier studies carried out in Tunisia [6,8] and worldwide [2,3,4] that showeda dissemination of *bla*_VIM-2_. Interestingly, we report here for the first time in North Africa the emergence of *bla*_GES-5_ and *bla*_GES-9_ harboring CPPA isolates thathad been reported in European [18,19], Asian [20], South African [21], and South American [22] studies.

The high levels of resistance among isolated *P. aeruginosa*, especially in ICUs where there are critically ill patients who underwent invasive procedures using multiple devices and broader spectrum antibiotics, emphasizes the need for measures to prevent the clinical dissemination of these isolates. A similar scenario was described by Koutsogiannou et al., who also reported the clonal dissemination of MDR *P. aeruginosa* in a university hospital [23].

## 5. Conclusions

Carbapenemase enzymes among *P. aeruginosa* isolates in Tunisia remain poorly investigated. This study reveals new information about carbapenemase enzymes among *P. aeruginosa* isolates in Tunisia by demonstrating the first detection of of GES-5 and GES-9 carbapenemases as well as the co-occurrence of *bla*_GES-5_ and *bla*_VIM-11_ in one isolate. Moreover, the CIMTris is a promising approach for detecting carbapenemase activity in *P. aeruginosa* and merits further testing. These findings are of great concern because the rapid dissemination of MDR strains represents a major therapeutic and epidemiological threat and requires the implementation of strict hygiene procedures and regular surveillance studies.

## Figures and Tables

**Figure 1 antibiotics-11-00858-f001:**
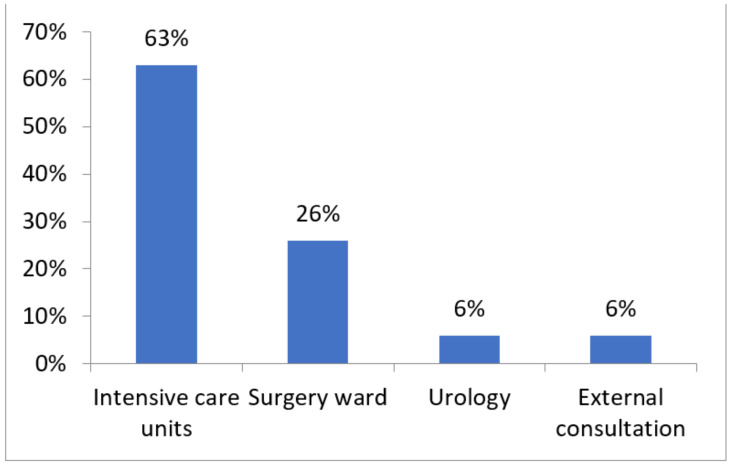
Distribution of *P. aeruginosa* isolates according to wards.

**Figure 2 antibiotics-11-00858-f002:**
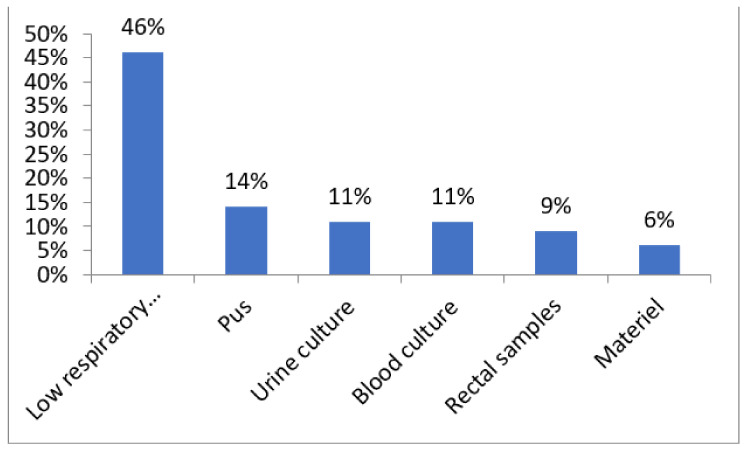
Distribution of *P.aeruginosa* isolates according to sample types.

**Figure 3 antibiotics-11-00858-f003:**
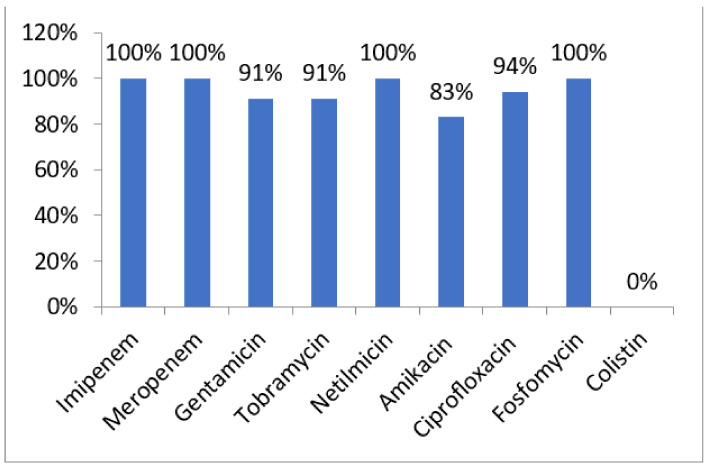
Antibiotic resistance rates among *P.aeruginosa* isolates.

**Table 1 antibiotics-11-00858-t001:** Oligonucleotides used in this study.

Gene	Primer ^a^	Sequence (5′–3′) ^c^	Product Size (bp) ^b^	Reference
** *bla* _GES_ **	MultiGES-F	AGTCGGCTAGACCGGAAG	399	[14]
	MultiGES-R	TTTGTCCGTGCTCAGGAT		
** *bla* _OXA-48_ **	MultiOXA-48 F	GCTTGATCGCCCTCGATT	281	[14]
	MultiOXA-48 R	GATTTGCTCCGTGGCCGAAA		
** *bla* _IMP_ **	MltiIMP-F	TTGACACTCCATTTACDG	232	[15]
	MultiIMP-R	GATYGAGAATTAAGCCACYCT		
** *bla* _VIM_ **	MultiVIM-F	GATGGTGTTTGGTCGCATA	390	[15]
	MultiVIM-R	CGAATGCGCAGCACCAG		
** *bla* _KPC_ **	MultiKPC-F	CATTCAAGGGCTTTCTTGCT C	798	[15]
	MultiKPC-R	ACGACGGCATAGTCATTTGC		
** *bla* _BIC_ **	MultiBIC-F	TATGCAGCTCCTTTAAAGGGC	537	[15]
	MultiBIC-R	TCATTGGCGGTGCCGTACAC		
** *bla* _NDM_ **	MultiNDM-F	GGTTTGGCGATCTGGTTTTC	621	[15]
	MultiNDM-R	CGGAATGGCTCATCACGATC		
** *bla* _AIM_ **	MultiAIM-F	CTGAAGGTGTACGGAAACAC	322	[15]
	MultiAIM-R	GTTCGGCCACCTCGAATTG		
** *bla* _GIM_ **	MultiGIM-F	TCGACACACCTTGGTCTGAA	477	[15]
	MultiGIM-R	AACTTCCAACTTTGCCATGC		
** *bla* _SIM_ **	MultiSIM-F	TACAAGGGATTCGGCATCG	570	[15]
	MultiSIM-R	TAATGGCCTGTTCCCATGTG		
** *bla* _DIM_ **	MultiDIM-F	GCTTGTCTTCGCTTGCTAACG	699	[15]
	MultiDIM-R	CGTTCGGCTGGATTGATTTG		
** *bla* _SPM_ **	SPM-F	AAAATCTGGGTACGCAAACG	271	[15]
	SPM-R	ACATTATCCGCTGGAACAGG		

^a^ F, sense primer; R, antisense primer. ^b^ Nucleotide numbering begins at the initiation codons of genes. ^c^ D = A or G or T; Y = C or T.

**Table 2 antibiotics-11-00858-t002:** Characteristics of imipenem- and ceftazidime-resistant *P. aeruginosa* isolates (*n* = 35).

Strains	Specimen	Ward	Date of Isolation (Day/Month/Year)	Minimal Inhibitory Concentration (µg/mL)			Resistance to Non β-lactams	Phenotypic Detection of Carbapenemases			*bla* Genes
				Imipenem (4–8) *	Meropenem (2–8) *	Colistin (4) *		mCIM	CIMTris	mHodge	
S_1_	Pus	Urology	21 August 2014	512	128	4	GEN, AMN, NET, TOB, CIP, FOS	-	-	-	-
S_2_	Urine	ICU	21 August 2014	16	4	4	GEN, AMN, NET, TOB, CIP, FOS	-	+	-	*bla* _VIM-2_
S_3_	Pulmonary	ICU	21 August 2014	32	16	4	GEN, AMN, NET, TOB, CIP, FOS	-	+	-	*bla* _VIM-2_
S_4_	Pulmonary	ICU	30 August 2014	512	128	4	GEN, AMN, NET, TOB, CIP, FOS	+	+	+	*bla* _VIM-1_
S_5_	Pulmonary	ICU	8 September 2014	32	16	1	GEN, AMN, NET, TOB, CIP, FOS	-	+	-	*bla* _GES-9_
S_6_	Material	Surgery	10 September 2014	16	8	2	GEN, AMN, NET, TOB, CIP, FOS	-	+	-	*bla* _GES-9_
S_7_	Pulmonary	ICU	23 September 2014	8	4	4	GEN, AMN, NET, TOB, CIP, FOS	-	+	-	*bla* _GES-5_
S_8_	Pulmonary	Surgery	30 October 2014	16	8	2	GEN, AMN, NET, TOB, CIP, FOS	-	+	-	*bla* _GES-5_
S_9_	Pulmonary	ICU	24 December 2014	512	256	1	GEN, AMN, NET, TOB, CIP, FOS	+	+	-	*bla* _VIM-2_
S_10_	Pulmonary	ICU	20 December 2014	8	4	1	GEN, NET, TOB, CIP, FOS	-	+	-	*bla* _GES-5_
S_11_	Urine	EC	06 February 2015	8	8	4	GEN, AMN, NET, TOB, CIP, FOS	-	-	-	-
S_12_	Pulmonary	Surgery	27 June 2015	16	16	4	GEN, AMN, NET, TOB, CIP, FOS	-	+	-	-
S_13_	Urine	EC	30 June 2015	16	8	4	GEN, NET, TOB, CIP, FOS	-	+	-	-
S_14_	Pulmonary	Surgery	19 September 2015	8	8	2	GEN, AMN, NET, TOB, CIP, FOS	-	+	-	*bla* _GES-5_
S_15_	Pus	Surgery	30 September 2015	128	128	4	GEN, AMN, NET, TOB, CIP, FOS	-	+	-	*bla* _GES-5_ *, bla* _VIM-11_
S_16_	Pus	ICU	18 March 2016	16	8	2	GEN, NET, TOB, CIP, FOS	+	+	+	*bla* _VIM-2_
S_17_	Pulmonary	ICU	06 April 2016	16	8	2	AMN, NET, TOB, CIP, FOS	-	+	-	*bla* _GES-5_
S_18_	Pus	ICU	26 May 2016	32	32	16	GEN, AMN, NET, TOB, CIP, FOS	-	+	-	*bla* _GES-5_
S_19_	Blood	ICU	13 June 2016	4	4	2	GEN, AMN, NET, TOB, FOS	+	+	-	*bla* _VIM-2_
S_20_	Blood	ICU	12 July 2016	32	8	2	GEN, AMN, NET, TOB, CIP, FOS	-	+	-	*bla*_GES-5_, *bla*_VIM-2_
S_21_	Blood	Surgery	18 August 2016	32	16	2	AMN, NET, TOB, CIP, FOS	-	+	-	*bla* _GES-5_
S_22_	Blood	ICU	16 August 2016	32	8	4	GEN, AMN, NET, TOB, CIP, FOS	-	+	-	*bla* _GES-5_
S_23_	Pulmonary	Surgery	21 August 2016	16	8	4	AMN, NET, TOB, CIP, FOS	+	+	-	*bla* _GES-5_
S_24_	Puncture	Surgery	13 October 2016	16	8	1	GEN, AMN, NET, TOB, CIP, FOS	-	+	-	*bla* _GES-5_
S_25_	Pulmonary	Surgery	31 October 2016	16	8	2	GEN, AMN, NET, TOB, CIP, FOS	+	+	-	*bla* _GES-5_
S_26_	Material	ICU	25 October 2016	16	8	2	GEN, AMN, NET, TOB, CIP, FOS	-	-	-	-
S_27_	Pulmonary	ICU	15 February 2017	128	64	1	AMN, NET, TOB, CIP, FOS	+	+	-	*bla* _VIM-2_
S_28_	Pus	ICU	17 February 2017	16	8	1	GEN, AMN, NET, TOB, CIP, FOS	-	+	-	-
S_29_	Pulmonary	ICU	24 June 2017	16	8	1	GEN, AMN, NET, TOB, CIP, FOS	-	-	-	-
S_30_	Urine	Urology	28 August 2017	16	8	2	GEN, AMN, NET, TOB, CIP, FOS	-	+	-	-
S_31_	Rectal	ICU	16 September 2017	16	8	1	GEN, AMN, NET, TOB, CIP, FOS	+	+	-	*bla* _GES-5_
S_32_	Pulmonary	ICU	09 October 2017	8	4	1	GEN, AMN, NET, TOB, CIP, FOS	+	+	-	*bla* _VIM-2_
S_33_	Rectal	ICU	09 October 2017	32	32	4	GEN, AMN, NET, TOB, CIP, FOS	+	+	+	*bla* _VIM-2_
S_34_	Rectal	ICU	22 November 2017	64	32	2	GEN, AMN, NET, TOB, CIP, FOS	+	+	-	*bla* _GES-5_
S_35_	Pulmonary	ICU	30 November 2017	32	16	2	GEN, AMN, NET, TOB, CIP, FOS	+	+	-	*bla* _GES-5_

EC: External consultation; ICU: Intensive care unit; *: MICs interpretive standard; mCIM: modified carbapenem inactivation method; CIMTris: carbapenem inactivation method Tris; mHodge test: modified Hodge test; +: Positive test; -: Negative; GEN: gentamicin; TOB: tobramycin; AMN: amikacin; NET: netilmicin; CIP: ciprofloxacin; FOS: Fosfomycin.

**Table 3 antibiotics-11-00858-t003:** Comparison of three phenotypic methods for carbapenemase detection in *Pseudomonas aeruginosa* strains.

	mCIM	CIMTris	mHodge Test
True positive	12	28	3
True negative	8	4	8
False positive	0	3	0
False negative	15	0	24
Specificity (%)	100	90,4	100
Sensitivity (%)	34.8	100	25

mCIM: modified carbapenem inactivation method; CIMTris: carbapenem inactivation method Tris; mHodge test: modified Hodge test; %: percentage.

**Table 4 antibiotics-11-00858-t004:** Comparison of three phenotypic methods for carbapenemase detection in *Pseudomonas aeruginosa* strains according to carbapenemase encoding genes.

Phenotypic Tests		PCR Results		
		Carbapenemase Coding Genes	*bla* _VIM_	*bla* _GES_
mHodge test	Positive (*n*)		3	1
	Negative (*n*)		24	15
	Sensitivity (%)	12.5	30	5
	Specificity (%)	100	100	86.66
mCIM	Positive (*n*)		7	5
	Negative (*n*)		20	10
	Sensitivity (%)	46.15	63.7	26.31
	Specificity (%)	100	83.33	62.5
CIMTris	Positive (*n*)		9	15
	Negative (*n*)		7	5
	Sensitivity (%)	96.15	81.81	74
	Specificity (%)	44.44	29	21.42

mCIM: modified carbapenem inactivation method; mHodge test: modified Hodge test; CIMTris: carbapenem inactivation method Tris.

## Data Availability

Not applicable.

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
