# Peer review of "Evaluation of Three Carbapenemase-Phenotypic Detection Methods and Emergence of Diverse VIM and GES Variants among Pseudomonas aeruginosa Isolates in Tunisia"

_antibiotics, 2022, doi:10.3390/antibiotics11070858_

Round 1

Reviewer 1 Report

The manuscript is a straight forward description of Pseudomonas aeruginosa clinical isolates from Tunisia that were assessed for antibioitic resistnace and screened for the presence of resistance genes by PCR.  

Only minor revisiosn are requested by this reviewer. See below for specific comments:

Specific comments:

Line 36: change "in oprD gene" to "in the oprD gene"

Lines 39-43 need to have references added within

Line 51-52 needs a reference

Line 84: provide name of the commercialized kit

Lines 113-115: how many isolates had more than one resistance gene deteced by PCR?

Author Response

Response to Reviewer 1 Comments

Specific comments:

Line 36: change "in oprD gene" to "in the oprD gene"

Response 1: The term “in oprD gene" was changed by "in the oprD gene": see line 38

Lines 39-43 need to have references added within

Response 2: The reference was added see line 46 ([3]: Kateete et al, 2016)

Line 51-52 needs a reference

Response 3: The reference was added see line: 54 ([10]: Uechi, K et al, 2017)

Line 84: provide name of the commercialized kit

Response 4: The name of the kit was alredy provided, UMIC, I have add the region and the country (UMIC, Biocentric, Bandol-France): see line 85

Lines 113-115: how many isolates had more than one resistance gene deteced by PCR?

Response 5: there is two isolates that had more than one resistance gene deteced by PCR, this was corrected in the text: see line 147

Reviewer 2 Report

In this version, three methods are compared to detect carbapenem resistant in Pseudomonas aeruginosa

The following comments are made

1. Line 70. Mention Table 2

2. Line 130. Figure Footer Table 2: It can't see where the letters a, b, c, are, and PFGE that needs to be removed.

3. Line 139. Figure caption does not mention CIMTris

4. Line 149. Figure caption must be CIMTris, not mCIM

5.Line 103-117. It would be better explained with bar graphs.

6. In the Discussion the differences found between its three techniques, Table 3 and Table 4, are not discussed. Which is the best and why? Which do you recommend and why?

7. What is the conclusion regarding the three methods evaluated?

8. References 12, 16. The journal name is abbreviated.

Author Response

Response to Reviewer 2 Comments

In this version, three methods are compared to detect carbapenem resistant in Pseudomonas aeruginosa

The following comments are made

  1. Line 70. Mention Table 2

Response 1: Table 2 was added: see line 72

  1. Line 130. Figure Footer Table 2: It can't see where the letters a, b, c, are, and PFGE that needs to be removed.

Response 2: a, b, c, are, and PFGE were removed from figure footer  of Table 2 as recommanded

  1. Line 139. Figure caption does not mention CIMTris

Response 3: CIMTris was aded in Figure caption as recommanded

  1. Line 149. Figure caption must be CIMTris, not mCIM

Response 4:  Figure caption was corrected

  1. Line 103-117. It would be better explained with bar graphs.

Response 5: Bar graphs was added: Figure 1,2 and 3. See lines 123-137

  1. In the Discussion the differences found between its three techniques, Table 3 and Table 4, are not discussed. Which is the best and why? Which do you recommend and why?

Response 6: A discussion on the differences found between its three techniques was added: see lines 200-208

  1. What is the conclusion regarding the three methods evaluated?

Response 7: see lines 208-211

  1. References 12, 16. The journal name is abbreviated.

Response 8: The journal name of references 12 and 16 was corrected

Round 2

Reviewer 2 Report

The authors made the suggested changes